# How do we classify organ involvement in Chagas disease? A systematic review of organ involvement since 1909, Highlighting the urgent need for a universal classification system in Chronic Chagas disease

**Irene Losada Galván**[1,2,3]*, **Magdalena García**[4], **Alejandro Marcel Hasslocher-Moreno**[5], **Ariadna Ortiga**[1], **Sergi Sanz**[1,6,7], **Israel Molina**[8,9], **Joaquim Gascón**[1,9], **Maria-Jesus Pinazo**[2,9,10]

**1** ISGlobal, Barcelona, Spain, **2** Facultat de Medicina i Ciències de la Salut, Universitat de Barcelona (UB), Barcelona, Spain, **3** Hospital Universitario 12 de Octubre, Madrid, Spain, **4** Hospital General Universitario de Valencia, Valencia, Spain, **5** Evandro Chagas National Institute of Infectious Diseases, Oswaldo Cruz Foundation, Brazil, **6** Department of Basic Clinical Practice, Faculty of Medicine, University of Barcelona, **7** CIBER Epidemiología y Salud Pública (CIBERESP), Instituto de Salud Carlos III, Spain, **8** International Health Unit Vall d'Hebron-Drassanes, Infectious Diseases Department, Vall d'Hebron University Hospital, PROSICS Barcelona, Barcelona, Spain, **9** CIBER de Enfermedades Infecciosas, Instituto de Salud Carlos III (CIBERINFEC, ISCIII), Spain, **10** Drugs for Neglected Diseases initiative, Rio de Janeiro, Brasil

\* irenelosada@gmail.com, irene.losada@isglobal.org

## Abstract

Chagas disease (CD) is recognized as one of the 20 neglected tropical diseases by the World Health Organization (WHO), posing a significant global health challenge. The objective of this work was to conduct a systematic methodology review to explore the different classifications used to describe the presence and degree of organ involvement in patients with CD since the disease's description in 1909. We searched relevant electronic medical databases from their inception dates to July 2023. We also delved into historical variations and revisions of each classification, the necessary diagnostic methods, their prognostic value, and their uptake. Our study underscores the conspicuous absence of a universally accepted CD classification system for cardiac and digestive involvement, both in the context of clinical trials and within current clinical guidelines. This endeavour will facilitate cross-population comparisons if clinical manifestations and complementary test results are available for each patient, constituting a pivotal stride toward identifying precise prognoses and establishing a minimum data set requisite for a fitting CD classification, tailored to the test availability in both endemic and non-endemic regions.

## Author summary

Chagas disease (CD) is a serious global health issue, and in our research, we aimed to investigate how doctors classify the impact of this disease on different organs. CD is one

**Data Availability Statement:** The authors confirm that all data underlying the findings are fully available without restriction. All relevant data are within the paper and its Supporting information files.

**Funding:** We acknowledge support from the grant CEX2023-0001290-S funded by MCIN/AEI/ 10.13039/501100011033, and support from the Generalitat de Catalunya through the CERCA Program. The funders had no role in study design, data collection and analysis, decision to publish, or preparation of the manuscript.

**Competing interests:** The authors have declared that no competing interests exist.

of the neglected tropical diseases recognized by the World Health Organization. It is important for us to understand how this disease affects people because it can lead to severe health problems.

To conduct our study, we reviewed the various ways doctors have classified CD since it was first described in 1909. We looked at how these classifications have changed over time, the tests used to diagnose CD, and how these classifications are applied in clinical practice and research.

One crucial discovery we made during our research is that there isn't a universally accepted classification system for CD, especially when it comes to assessing its impact on the heart and digestive system. This absence of a standard classification system makes it difficult to compare CD cases across different regions and to predict the disease's progression in individual patients.

We believe that establishing a standardized classification system for CD is of utmost importance. Such a system would greatly assist doctors and researchers in gaining a better understanding of the disease, making more accurate predictions about how it will affect patients, and improving CD diagnosis and treatment, both in regions where the disease is prevalent and in areas where it's not.

In summary, our study highlights the urgent need for a standardized method to classify Chagas disease, which is a significant global health concern. Establishing a common classification system would simplify the study and treatment of the disease, benefiting people worldwide.

## Introduction

Chagas disease (CD), recognized as one of the 20 neglected tropical diseases by the World Health Organization (WHO) [1], poses a significant global health challenge. With an estimated 70 million people at risk and approximately six million currently infected, the disease accounts for about 14,000 deaths annually worldwide [2,3]. Although many individuals with the infection remain asymptomatic and display no signs of organ involvement, around 30–40% of cases may develop cardiac or digestive complications later in life, usually a decade or more after they acquire the infection.

Predicting the individual probability of disease progression remains a challenge, as there are currently no reliable tools available for this purpose. Cardiological manifestations, affecting 14–45% [4] of patients during their lifetime, encompass various heart-related issues, including electrocardiographic changes represented by the association of left anterior hemiblock with complete right bundle branch block, and the presence of isolated premature contractions, in addition to of echocardiographic changes represented by segmental changes in myocardial contractility and left ventricular systolic dysfunction. Advanced stages of cardiomyopathy can lead to complete atrioventricular block, ventricular tachycardia and severe myocardial dysfunction, culminating in severe bradycardia, sudden death and refractory heart failure and/or thromboembolism [5].

Digestive symptoms, affecting 15–20% of patients, arise from changes in the enteric nerve plexus, causing motility, secretion, and absorption issues in the digestive system [6,7]. The most frequent presentations are megacolon and megaesophagus, which significantly impact patients' quality of life [8]. Additionally, there is a 'mixed' form of involvement, affecting 5–20% of cardiomyopathy patients, and less common manifestations include thromboembolic phenomena with neurological involvement [9].

There are several clinical classifications for the visceral involvement of CD, but five of them stand out, all staging Chagas disease with a focus on heart disease: Modified Los Andes [10], Kuschnir [11], 2nd Brazilian consensus [12], I Latin American guidelines [13] and Rassi [14]. All these classifications, attempt to stratify patient's prognosis based on different information such as ECG, chest X-ray, echocardiogram, and clinical heart failure symptoms for cardiomyopathy, among others. For gastrointestinal involvement, oesophageal and colonic alterations are mainly classified based on radiological findings [15].

Historically, estimating the prevalence of chronically affected individuals has been a challenge as a consequence of being a neglected disease. However, recent initiatives such as the establishment of *T. cruzi* infection as a notifiable disease in some countries [16–19], have provided more precise estimates. Yet, there remains a lack of knowledge concerning the extent of visceral involvement in chronically infected individuals. One reason for this is that there is no globally accepted classification used extensively for organ involvement.

Having a well-defined map of the various classifications and required variables is crucial for establishing comparisons among previous studies and analysing results from patient cohorts that employed different classifications. This initial step is essential in the search for prognostic factors that could be assessed through clinical parameters.

## Methods

The objective of this work was to conduct a systematic methodology review to explore the different classifications used to describe the presence and degree of organ involvement in patients with chronic CD since the disease's description in 1909. We analysed the diagnostic methods required for each classification, the criteria for inclusion or cut-off points, and their current level of use and acceptance. Additionally, we delved into historical variations and revisions of each classification, the necessary diagnostic methods, their prognostic value, and their uptake. This will also allow comparisons between cohorts initially classified by different methods.

### Electronic literature search

We performed a literature review using a systematic search. A comprehensive search strategy containing combinations of terms including and relating to 'Chagas disease', 'Trypanosoma cruzi' and 'classification', using a combination of medical subject headings (MeSH) and freetext terms wherever relevant and possible.

Electronic medical databases were searched for potential studies from their inception dates to July 2023, and included: PubMed/MEDLINE (1964), clinicaltrials.gov (2000), Cochrane (1993), Scopus (1945), EMBASE (1947), LILACS (1982) and Web of Science (1945). The Search Engine is available in S1 Table.

### Uptake of classifications

A separate search was performed to determine the acceptability of each classification. On the one hand, a query was made for each individual article to determine the number of bibliographic citations for each one, using Google Scholar [20], PubMed [21], and Dimensions [22]. The average number of citations from each of the available sources was used to obtain an approximation of the popularity or acceptance of each classification.

We also searched the clinicaltrials.gov [23] database in order to assess which classification or classifications were used as a reference in each study in CD. Finally, national guidelines from endemic and non-endemic countries were consulted to assess the current popularity of each classification.

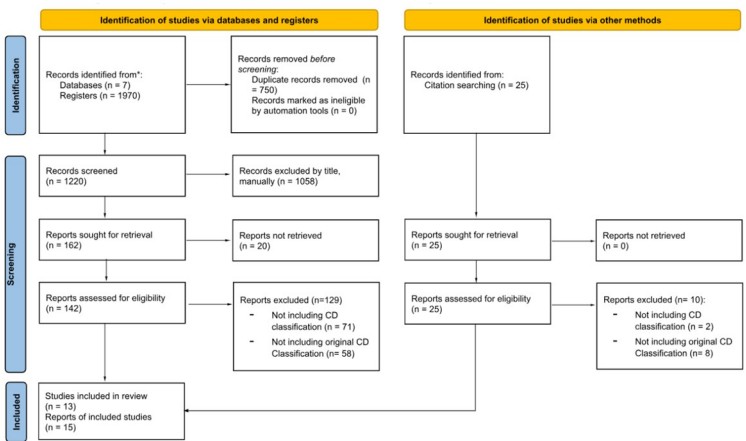

**Fig 1. Flowchart of study selection.**

## Results

After eliminating duplicates, a total of 1219 titles were reviewed, discarding 1058 of them, mainly because they were non-human research, and the remainder because they did not contain original classifications of Chagas disease. Of the 161 papers selected, 20 of them could not be found despite specific requests to the originating libraries. Moreover, 25 new references were added through bibliographic citation review. Of a total of 166 references evaluated, 73 did not include Chagas classification and 66 included non-original classifications. Therefore, 27 articles relevant to the research question were selected. A Prisma flow of the reviewed articles can be found in Fig 1. Flowchart of study selection. Among the 27 selected articles, eight contain digestive and 22 cardiological classifications, including three articles containing both; comprising different parameters as shown in Fig 2. The detailed distribution of each parameter according to the different classifications can be found in S2 Table.

### Cardiological classifications

**Carlos Chagas** himself, in **1916**, only seven years after his first account of the disease, described in detail the acute form and its possible -not excluding- presentations (fever, myxoedema, steatosis, myocarditis, meningo-encephalitis) as well as the evidence of involvement of

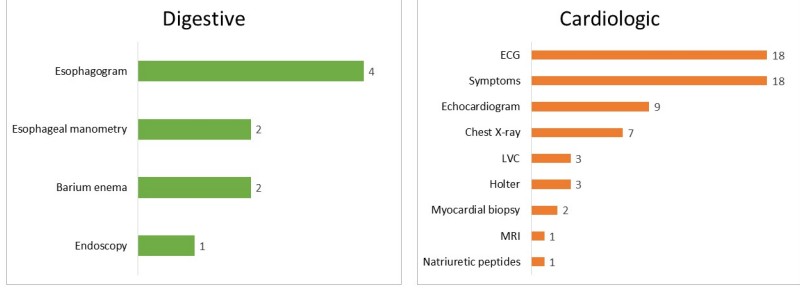

**Fig 2. Parameters included in cardiac (n = 22) and digestive (n = 8) classifications.** ECG: electrocardiogram; LVC: left ventricular cineangiography; MRI: magnetic resonance imaging.

other systems generally not clinically apparent (muscular, cutaneous and even genital) [24]. In the chronic phase, he described: (i) cardiac form, essentially muscular, with either progressive heart failure or sudden death; (ii) nervous form: motor, cognitive and speech disorders; (iii) adrenal: glandular insufficiency leading to "aggravation of the asthenia determined by the weakness of the myocardium". He also described other glandular dystrophies such as hypothyroidism and, especially, pluriglandular infantilism (he postulates infection as an agent of endemic goiter) that he thought related to CD. Finally, he outlined (iv) an indeterminate chronic form in which, contrary to the current definition, there may be clinical manifestations but there is no predominance of involvement of a specific organ or system. Therefore, they would not necessarily be disease-free individuals.

In **1966**, a study of two populations highly endemic for CD in **Venezuela** classified patients according to the presence or absence of heart disease based on electrocardiographic criteria [25]: (i) ventricular repolarization disorders (T wave alterations and S-T segment alterations), (ii) Intraventricular conduction disorders: complete or incomplete right or left bundle branch block or unclassified intraventricular block, (iii) cardiac rhythm disorders: supra or ventricular extrasystoles, (iv) Low or High voltage, (v) Right or left ventricular enlargement, (vi) P-wave alterations, (vii) Inactive electrical zones: abnormal Q waves or absence of precordial R and (viii) AV conduction disorders. Those patients with alterations compatible with cardiac involvement, were classified further according to their functional class between I and IV (no criteria provided).

In a two-volume work published in **1967** [26] and **1968** [27], Argentinean researchers explored cardiac involvement of Chagas disease, categorizing myocardial involvement into five types: (i) acute myocarditis, (ii) subacute myocarditis, (iii) chronic myocarditis with recoverable trypanosomes in blood, (iv) latent or indeterminate phase (potential cardiopaths), and (v) chronic myocarditis.

Within the chronic forms, chronic myocarditis with detectable trypanosomes in the blood was examined through two cases of symptomatic heart disease, confirmed by positive xenodiagnosis using puppy dogs, occurring years after the initial *T. cruzi* infection. The latent or indeterminate phase was discussed with the idea that once the balance between parasite and host is restored, the former chagasic individual may only show serological reactions indicating the existence of the infection throughout their life. Additionally, the progression of the disease was addressed, noting that the majority of chagasic individuals who were initially infected, even those who displayed signs of acute myocarditis, tend to recover clinically. However, after a period of 10 to 30 years, some of them may develop evident signs of chronic myocarditis.

This work established diagnostic criteria for chronic chagasic myocarditis, considering three different but interdependent problems: heart failure, electrocardiographic and radiographic disorders, and sudden death (Table 1). However, no criteria were established for determining different levels of involvement.

The **PAHO/WHO report of 1974** [28] is the first record of a comprehensive classification of clinical forms: cardiological, digestive, and neurological involvement, although the neurological form is not described in the paper because the authors state that the exact information needed is not yet available. Chronic heart involvement is classified according to symptoms, electrocardiographic and chest x-ray disorders, Table 2. It is noteworthy that patients with no symptoms or alterations in the complementary tests are included in the first degree of cardiac involvement. In terms of prognosis, it is stated that there is a risk of unexpected death in patients with grade 2 to 4 of disease.

Another review, the **Anatomo-clinical Chagas classification** [29], gathers anatomopathological findings through some examples of necropsies, establishing the following classification:

**Table 1. Diagnostic criteria for chronic chagasic myocarditis [26,27].**

| Minor criteria | Major criteria |
|---|---|
| "Cardiac irregularity": isolated ventricular extrasystoles, especially on exertion. | Marked exertional dyspnoea |
| Electrocardiographic abnormalities: isolated premature contractions, complete right bundle branch block, ventricular repolarization disorders. | Decubitus dyspnoea |
| Discrete global increase of the cardiac silhouette in chest X-ray. | Oedema |
| | signs of embolism |
| | multifocal extrasystoles |
| | conduction block, especially RBBB |
| | ventricular repolarization disturbances compatible with cardiac tip hypoxia and signs |

- Myopathic form: cardiomegaly, progressive heart failure, intense myocarditis on histology and with or without conduction disturbances on ECG.

- Neuropathic-vegetative form: some unspecified cardiological involvement, neurovegetative dyskinesias of the digestive tract, some unspecified gland involvement.

- Dromopathic: conduction abnormalities and/or automaticity on ECG. Discrete myocardial inflammation.

- Mixed form.

Possible involvement of the central and peripheral sympathetic nervous systems and skeletal musculature is also described. However, there are no clearly defined criteria for each group and there are no subsequent studies confirming the findings in larger samples or different centres.

In 1982 [30] a modification of the **Minnesota Code** [31] to standardize recording of electrocardiographic data in population-based studies was presented. In summary, changes are proposed under the headings of Q and QS pattern, QRS axis, high-amplitude R waves, ST junction and segment depression, T-wave items, AV conduction defect, ventricular conduction defect, arrhythmias and miscellaneous items at rest. This work proposed changes in the nomenclature, such as different cut-off points for some measurements, to better characterize possible electrocardiographic changes due to CD in children and adolescents. However, no specific changes of the disease or staging criteria are defined. Moreover, the updated 2010 Minnesota Code [32] does not include any of the proposed changes.

**Table 2. Classification of chronic cardiac involvement according to PAHO/WHO report of 1974 [28].**

| Degree | Symptoms | ECG | Chest x-ray* |
|---|---|---|---|
| 1st | none | normal | normal |
| 2nd | moderate or none | - complete right bundle branch block<br>- ventricular repolarization disorders<br>- ventricular extrasystoles | normal or mild hypertrophy |
| 3th | evident | - complete right bundle branch block with left axis deviation<br>- electrically inactive zones | moderate hypertrophy |
| 4th | very pronounced with heart failure | - severe and multiple disturbances<br>- complex and severe arrhythmias<br>- extensive electrically inactive areas | extreme cardiomegaly |

ECG: electrocardiogram; * No criteria are provided to differentiate between radiologic degrees of cardiomegaly.

**Table 3. Kuschnir [11].**

|   | Symptoms | ECG | Chest x-ray |
|---|---|---|---|
| 0 | asymptomatic | normal | normal |
| I | asymptomatic | abnormal | normal |
| II | asymptomatic | abnormal | mild or moderate cardiomegaly |
| III | CHF symptoms | indifferent | indifferent |

ECG: electrocardiogram; CHF: chronic heart failure

**Table 4. 1982 Los Andes classification [10].**

|   |   | Symptoms | ECG | LVC |
|---|---|---|---|---|
| IA | No heart disease | Asymptomatic | normal | normal |
| IB | Early myocardial damage | Asymptomatic | normal | abnormal |
| II | Advanced myocardial damage | No HF | abnormal | abnormal |
| III | Congestive Cardiomyopathy | CHF symptoms | abnormal | abnormal |

ECG: electrocardiogram; LVC: left ventricular cineangiography; CHF: chronic heart failure

In a 1985 [11] study on the assessment of cardiac function by radioisotope angiography in patients with chronic chagasic heart disease, **Kuschnir** et al. staged patients according to clinical criteria and ECG and chest radiographic abnormalities (Table 3). These groups showed a good correlation with functionality as measured by isotopically determined ejection fraction.

A series of three publications between 1982 and 1987 by the Universidad de **Los Andes** proposed a classification into stages of cardiac involvement known as the 'Los Andes' classification. Essentially very similar to each other, with some variations as described below.

The 1982 classification [10] considered four groups based on ECG, clinical features and the invasive left ventricular cineangiography (LVC) procedure, Table 4. The electrocardiographic alterations considered were arrhythmias, conduction defects and fibrotic areas.

In 1985 [33], the same researchers analysed the evolution of the groups in the previous publication. Groups IA and IB had a similar life expectancy to the control group in 10 years of follow-up, while patients in groups II and III showed a significant reduction in life expectancy.

In 1987 [34], a new classification was published which, in addition to the previous parameters, included the result of endomyocardial biopsies and introduced slight changes in the groups and in the meaning of each one, Table 5. In addition, the following electrocardiogram abnormalities were considered normal for the purposes of this classification: incomplete bundle branch block, first-degree atrioventricular block, nonspecific ST-T changes).

**Table 5. 1987 Los Andes Classification [34].**

|   |   | Symptoms | ECG | LVC | Biopsy |
|---|---|---|---|---|---|
| 0 | No myocardial damage | Asymptomatic | normal | normal | normal |
| IA | Subcellular myocardial damage | Asymptomatic | normal | normal | abnormal |
| IB | Segmental myocardial damage | Asymptomatic | normal | abnormal | abnormal |
| II | Advanced myocardial damage | No HF | abnormal | abnormal | abnormal |
| III | End-stage cardio-myopathy | HF symptoms | abnormal | abnormal | abnormal |

ECG: electrocardiogram; LVC: left ventricular cineangiography; HF: heart failure

**Table 6. 1995 Cardiology congress [35].**

|  |  | Symptoms | ECG | Chest x-ray |
|---|---|---|---|---|
| I | Subclinical or early | Asymptomatic | Normal | normal |
| IIA | Latent or undetermined* | No HF | Conduction disorders | CI < 0,55 |
| IIB |  |  | Ventricular arrhythmias |  |
| IIC |  |  | A + B |  |
| III | Advanced | HF symptoms | Pathologic | cardiomegaly: CI > 0,55 |

ECG: electrocardiogram; LVC: left ventricular cineangiography; HF: heart failure; CI: Cardiothoracic Index;

*The term 'latent or indeterminate' here includes patients who already have myocardial damage, contrary to the more widespread use of the term, which refers to patients without organic damage.

Therefore, Los Andes classifications require invasive methods such as ventriculography and endomyocardial biopsy. The ECG abnormalities are very general in the first classification, and somewhat more specific in the third.

The next attempt at classification was made at a **cardiology congress** in **1995** [35]. They simplified into 3 groups according to clinical, ECG and chest radiography, Table 6. Chagasic ECG findings are described generically as conduction disorders and ventricular arrhythmias, without further detail.

In a generic document published by the **WHO** in 1996 [36], aimed at the definition and classification of **cardiomyopathies in general**, a brief mention is made of chagasic cardiomyopathy within the 'specific' and 'inflammatory' cardiomyopathies. That is, it is classified among muscle diseases that are associated with specific cardiac or systemic disorders. And also, as an inflammatory cardiomyopathy which is defined by myocarditis in association with cardiac dysfunction. Myocarditis is an inflammatory disease of the myocardium and is diagnosed by established histological, immunological, and immunohistochemical criteria. This document does not address specific diagnostic criteria or stratification of chagasic cardiomyopathy.

Following the classification of chronic heart failure (CHF) published in 2001 by the **ACC/AHA** [37], from which CD was excluded, a 2005 study [38] aimed to evaluate the **prognostic performance** of this classification applied to a cohort of patients with CD, Table 7. In a 5-year follow-up, no progression to Chronic Chagas Cardiomyopathy (CCC) was observed among patients who had a normal ECG at baseline. Therefore, only patients with an abnormal ECG at baseline were included. Patients were divided into 4 groups according to the results of ECG,

**Table 7. 2005 application of ACA/AHA CHF classification to CCC [38].**

|  |  | Symptoms | ECG | Echocardiography |
|---|---|---|---|---|
| A | Altered ECG and normal ECHO | Asymptomatic | Abnormal | Normal |
| B | Abnormal ECHO and absent CHF |  |  |  |
| B1 | Normal function | Asymptomatic | Abnormal | LVEF > 55% |
| B2 | Mild dysfunction | Asymptomatic | Abnormal | LVEF 45–54% |
| B3 | Moderate dysfunction | Asymptomatic | Abnormal | LVEF 35–44% |
| B4 | Severe dysfunction | Asymptomatic | Abnormal | LVEF < 35% |
| C | Compensated CHF | Compensated CHF | Abnormal | Abnormal |
| D | Advanced CHF | Advanced CHF | Abnormal | Abnormal |

ECG: electrocardiogram; CHF: chronic heart failure; LVEF: left ventricular ejection fraction

**Table 8. Brazilian Consensus 2005 [39] and 2015 [12].**

|  | Symptoms | ECG | Echocardiography |
|---|---|---|---|
| A | Without CHF | Abnormal | Normal |
| B1 | Without CHF | Abnormal | LVEF > 45% |
| B2 | Without CHF | Abnormal | LVEF < 45% |
| C | Compensated CHF | Abnormal | Abnormal |
| D | Advanced CHF | Abnormal | Abnormal |

ECG: electrocardiogram; CHF: chronic heart failure; LVEF: left ventricular ejection fraction

echocardiogram, and clinical symptoms of heart failure. Survival curves showed a large difference between groups B and C, so group B was subdivided according to Left Ventricular Ejection Fraction (LVEF) measured by echocardiography; this was the first classification to include a non-invasive measurement of systolic function in Chagas.

A slightly modified version of this classification was adopted in the document known as the **First Brazilian Consensus** of 2005 [39]. It reduced the subgroups of group B with the cut-off point at LVEF 45%, Table 8. This document also contains non-original digestive classification related to the oesophagus, in agreement with that of Rezende [15,40].

A study of 71 individuals conducted in 2006 in Puebla [41], **Mexico**, divided patients into groups not commonly reported in previous or subsequent literature according to clinical and complementary examinations, Table 9. In summary, it is considered that any patient with symptoms such as dysphagia or constipation has a gastrointestinal involvement by CD (details in the following section). Regarding cardiac involvement, the presence of individuals with symptoms of CHF in the early stages is noteworthy.

An additional parameter in ECG evaluation is the **slope of the QRS**. A 2007 study [42] linked these slopes to the degree of myocardial damage in Chagas patients. This paper confirms the notion that both the up and down strokes of the QRS complex are steeper in healthy subjects. A stratification in groups according to ECG or Holter and echocardiogram results is used, not exactly the same as previous ones; Table 10. The aforementioned tendency in terms of QRS slopes is verified, but no specific cut-off points are established to elaborate a new classification. Moreover, there is no evidence of subsequent use of this parameter to classify organ involvement in CD.

In a study to determine the prognostic value of **natriuretic peptides** in Chagas disease [43], three clinical groups were established according to clinical (NYHA class), LVEF by TTE and ECG. Two additional groups consisted of *T.cruzi* negative patients with dilated cardiomyopathy due to other causes. It is noteworthy that group 1 included some patients with

**Table 9. Clinical forms Puebla 2006 [41].**

|  | Symptoms | ECG | Heart size* | Esophagogram / manometry | Barium enema |
|---|---|---|---|---|---|
| Indeterminate | Asymptomatic | normal | normal | normal | normal |
| Gastrointestinal | Dysphagia, constipation, retrosternal pain | normal | normal | megaoesophagus | megacolon |
| Mild Cardiopathy | NYHA I CHF symptoms | conduction disorders | normal | normal | normal |
| Moderate Cardiopathy | NYHA II-III CHF symptoms | bundle blocks | normal | normal | normal |
| Severe Cardiopathy | NYHA III CHF symptoms | altered | cardiomegaly | normal | normal |
| Combination form | Any combination of cardiomyopathy and digestive syndromes | | | | |

*by Chest x-ray and/or echocardiography; NHYA: New York Heart Association; CHF: chronic heart failure

**Table 10. Clinical groups for QRS slope evaluation [42].**

| | *T.cruzi* serology | ECG or Holter | Echocardiogram |
|---|---|---|---|
| 0 | - (non-reactive) | Normal | Normal |
| I | + (reactive) | Normal | Normal |
| II | + (reactive) | *Weak or moderate damage | Normal |
| III | + (reactive) | *Severe damage. Premature ventricular contractions or VT | Decreased LVEF |

*weak/moderate/severe damage criteria are not defined

VT: ventricular tachycardia; LVEF: left ventricular ejection fraction

electrocardiographic abnormalities, including pacemakers, Table 11. In any case, there was a tendency for higher natriuretic peptide levels with greater cardiac involvement, regardless of the cause of the marked cardiac dilatation. However, it does not establish a cut-off point that could contribute to improving the existing classifications.

In the 2010 review on CD by **Rassi (14)**, a classification is proposed that brings together most of the parameters described so far and adds the notion of increasing incidence of thromboembolic events and sudden death in more advanced stages. It also includes cardiologic symptoms that are not necessarily present in heart failure as well as NYHA classification of HF. Some of the headings are described in detail, such as electrocardiographic changes for each stage, while others describe only relative frequencies (rare, common, etc.) or staging of involvement without specifying cut-off points (e.g., mild cardiomegaly). Of note, it includes TTE abnormalities in the form of left ventricular wall motion abnormalities and the presence of ventricular aneurysms, but does not include LVEF. In Table 12 it is stated that there may be some minor changes in some features across the different stages, but it is not specified whether the trait of greater or lesser severity should prevail.

The first **Latin American guidelines** for the diagnosis and treatment of chagasic cardiomyopathy were published in 2011 [13], Table 13. The classification adopted maintains the staging nomenclature of the Brazilian consensus [39], which was based on Xavier's classification from 2005 [38]. However, there are significant changes with prognostic implications that can make the comparison between the two classifications confusing. On the one hand, in the Latin American guidelines, stage A describes patients in the indeterminate chronic phase by defining the ECG (and chest X-ray) as normal, whereas in the Brazilian consensus stage A, the ECG is abnormal. For TTE, the LVEF limit of 45% is removed and replaced by the subjective assessment of normal vs reduced. This significantly changes groups B1 and B2. This guideline considers patients without ventricular dysfunction (normal ejection fraction) as B1, whereas the Brazilian consensus considers patients with mild ventricular dysfunction (ejection fraction between 55% and 45%) as B1, in line with the survival curves observed in the Xavier's.

**Table 11. Clinical groups for natriuretic peptide evaluation.**

| | *T.cruzi* serology | Symptoms NYHA | ECG | Echocardiogram | ANP / BNP |
|---|---|---|---|---|---|
| 1 | + | Asymptomatic | normal/abnormal | LVEF > 50% | ↑ |
| 2 | + | I-II | abnormal | LVEF < 50% | ↑↑ |
| 3 | + | III-IV | abnormal | LVEF < 50% | ↑↑↑ |
| 4 | - | I-II | any | LVEF < 50% | ↑↑ |
| 5 | - | III-IV | any | LVEF < 50% | ↑↑↑ |

NHYA: New York Heart Association; ECG: electrocardiogram; LVEF: left ventricular ejection fraction; ANP:atrial natriuretic peptide; BNP: brain natriuretic peptide

**Table 12. Chronic Chagas Cardiomyopathy by Rassi [14].**

| | Cardiac symptoms* | NYHA class | ECG changes | Chest radiograph (cardiomegaly) | 24-h Holter (complex ventricular arrhythmias***) | 2D echocardiogram | | Thromboembolism | Sustained ventricular tachycardia | Sudden death |
|---|---|---|---|---|---|---|---|---|---|---|
| | | | | | | Left ventricular wall motion abnormalities | Left ventricular apical aneurysm | | | |
| Indeterminate | Absent | n/a | Absent | Absent | Very rare | Absent | Absent | Absent | Absent | Absent |
| Cardiac stage I | Absent or minimal | n/a | Non specific** | Absent | Rare | Rare | Very rare | Very rare | Rare | Rare |
| Cardiac stage II | Fairly common | I or II | RBBB with or without LAFB, monomorphic ventricular premature beats, diffuse ST-T changes, first or second degree AVB | Absent or mild | Common | Absent or segmental | Common | Fairly common | Common | Common |
| Cardiac stage III | Common | I, II or III | As for stage II plus Q waves, polymorphic ventricular premature beats, advanced AVB, severe bradycardia, low QRS voltage | Mild to moderate | Very common | Segmental or diffuse (mild to moderate) | Common | Fairly common | Common | Common |
| Cardiac stage IV | Common | II, III or IV | As for stage III plus atrial flutter or AF | Moderate to severe | Very common | Diffuse (severe) | Fairly common | Common | Fairly common | Fairly common |

NYHA: New York Heart Association. ECG: electrocardiogram. LAFB: Upper anterior fascicular block; AVB: Atrioventricular block;; RBBB: Complete right bundle branch block; AF: Atrial fibrillation; n/a: not applicable.

*Cardiac symptoms: Palpitations, presyncope, syncope, atypical chest pain,fatigue, and oedema.

**Non-specific ECG changes: Incomplete right bundle branch block, incomplete left anterior fascicular block, mild bradycardia, minor increase in PR interval, minor ST-T changes.

***Couplets or episodes of non-sustained ventricular tachycardia, or both.

**Table 13. 1st Latin American Guidelines.**

|     | Symptoms | ECG | Echocardiography |
|-----|----------|-----|------------------|
| A* | Without CHF | **Normal** | Normal |
| B1 | Without CHF | Abnormal | **Abnormal, but normal LVEF** |
| B2 | Without CHF | Abnormal | **Reduced LVEF** |
| C | **NYHA I, II or III** | Abnormal | Abnormal |
| D | **NYHA IV** | Abnormal | Abnormal |

*Normal Chest x-ray is also required

Conversely, in the clinical section, the previous descriptions are replaced by NYHA stages (with no change in the meaning of each).

Ten years after the first one, the **Second Brazilian Consensus** on Chagas Disease was published in 2016 [12]. In it, the classification in stages of cardiomyopathy was reissued, without changes, Table 8. It is worth noting that this edition includes an exhaustive list of the electro-cardiographic abnormalities that are considered significant for considering the ECG to be altered, Table 14. This document also contains non-original digestive classification related to the oesophagus, in agreement with that of Rezende and unchanged from 2005.

In 2023, the Brazilian Society of Cardiology issued guidelines regarding the diagnosis and treatment of Chagas cardiomyopathy patients [44]. Although these guidelines remain largely consistent with the initial Latin American guidelines [13], they introduce additional parameters for assessment. Notably, they include cardiomegaly observed on chest X-rays, the presence of complex ventricular arrhythmias detected via Holter monitoring, and for the first time, incorporate cardiac magnetic resonance data into the classification of CCC. Table 15.

## Digestive classifications

**Rezende** described in 1960 different degrees of oesophagopathy, Table 16 [15], in an article dedicated to oesophageal aperistalsis in general, in which he also describes in detail the radio-logical technique to promote reproducibility. Although the classification of oesophagopathy is

**Table 14. 2nd Brazilian consensus: Conventional ECG for evaluation of individuals with CD [12].**

|                      | Significant non-specific changes | ECG changes suggestive of CCC |
|----------------------|----------------------------------|-------------------------------|
| Frequency and rhythm | Sinus bradycardia (Heart rate > 40 bpm) | Heart rate < 40 bpm |
|                      |                                  | Sinus node dysfunction |
|                      |                                  | AF: Atrial fibrillation |
|                      |                                  | Isolated monomorphic Ventricular extrasystoles (VES) |
|                      |                                  | Polymorph or repetitive VES |
|                      |                                  | NSVT: Non-sustained ventricular tachycardia |
| Conduction           | First degree Atrioventricular block (AVB) | 2nd degree AVB |
|                      |                                  | Complete AVB |
|                      | IRBB: incomplete right bundle-branch block | RBBB: Complete right bundle branch block |
|                      | LAFB: Upper anterior fascicular block | LBBB: Left bundle branch block |
| Repolarization       | Non-specific ST-T changes | T changes |
| Others               | Low voltage | Electrically inactive areas |

**Table 15. Classification of the stages of chronic Chagas' disease.** Adapted from [44].

| | NYHA | ECG | Chest radiograph (cardiomegaly) | 24-h Holter (complex ventricular arrhythmias) | Echocardiography | | Myocardial fibrosis (late enhancement on cardiac MRI) |
|---|---|---|---|---|---|---|---|
| | | | | | segmental ventricular dysfunction | LVEF | |
| A | not applicable | Normal | Absent | Generally absent | Generally absent | ≥ 55% | May be present |
| B1 | I | Abnormal | Absent | May be present | May be present | ≥ 55% | Generally present |
| B2 | I | Abnormal | May be present | Generally present | May be present | <55% (usually 41–54%) | Generally present |
| C | I, II, III or IV | Abnormal | Generally present | Present | May be present | <55% (usually ≤ 40%) | Present |
| D | IV | Abnormal | Present | Present | May be present | Usually ≤ 25% | Present |

NHYA: New York Heart Association; ECG: lectrocardiogram; LVEF: left ventricular ejection fraction; MRI: magnetic resonance imaging

often cited with a 1982 article by Rezende himself [40], the classification is the same as the one he published in 1960.

This classification is based on the radiological examination of the oesophagus at different time points after ingestion of a radiological contrast solution. It considers: retention of contrast in the oesophagus, oesophageal calibre, contractile activity of the musculature, tonicity of the lower segment and elongation of the organ. As such, this classification has remained the standard for the classification of chagasic oesophagopathy.

The next allusion to digestive involvement in Chagas disease can be found in the aforementioned **PAHO/WHO report** of 1974 [28]. This report discusses the presence of oesophageal and colonic involvement. It specifically mentions megacolon as an indication of lower gastrointestinal involvement, suggesting a contrast enema as means of diagnosis. However, there is no clear establishment of diagnostic or stratification criteria from a digestive perspective.

Within the **anatomical-clinical classification** published by Curti in 1979 [29], a neuropathic-vegetative form is included. It describes a clinical spectrum from asymptomatic to neurovegetative dyskinesias of the digestive tract. However, it does not establish a classification of oesophageal or colonic involvement.

As for the colonic form, most classifications mention the presence or absence of megacolon. However, a **staging** of this **colonic involvement** was proposed by Silva in 2003 [45]. Using the same opaque enema technique, it is proposed to measure the transverse diameter of the colon

**Table 16. Rezende.** Aperistalsis of the oesophagus [15].

| | Radiological examination of the oesophagus after ingestion of radiological contrast solution |
|---|---|
| I | Oesophagus of apparently normal calibre on radiological examination. Slow transit. Slight contrast retention. |
| II | Oesophagus with small to moderate increase in calibre. Significant contrast retention. Tertiary waves are frequently observed, associated or not with hypertonia of the lower oesophagus. |
| III | Oesophagus with large increase in calibre. Hypotony of the lower oesophagus. Inapparent or reduced motor activity. Large contrast retention. |
| IV | Dolicomegaoesophagus. Oesophagus with great capacity of retention, atonic, elongated, folding over the diaphragmatic dome. |

**Table 17. Colonic involvement in CD [45].**

| Megacolon degree | Colonic diameter* |
|---|---|
| Grade 0 (normal) | 2–5 cm (incl) |
| Grade 1 | 5–9 cm (incl) |
| Grade 2 | 9–13 cm (incl) |
| Grade 3 | > 13 cm |

*transverse diameter of the colon at the height of the imaginary line joining the two iliac crests

**Table 18. Endoscopic classification of chagasic megaesophagus [46].**

| | Macroscopic aspect of the mucosa | Histology: submucosal vessels |
|---|---|---|
| A (mild esophagitis) | Normal | slightly faded |
| B (moderate esophagitis) | Pale | diminished |
| C (severe esophagitis) | Thick folds | not evident |

at the height of the imaginary line joining the two iliac crests. Thus, subjects would be divided into 4 groups as shown in Table 17.

In the aforementioned **Mexican** 2006 study [41], oesophageal involvement was evaluated by manometry and barium contrast studies. The presence of symptoms (with or without correlation in imaging tests), as well as changes in manometry compatible with gastroesophageal reflux were considered as criteria for digestive involvement. The rest of the criteria for digestive involvement coincide with other classifications in terms of motility alterations such as achalasia or visceral dilatation.

According to a 2012 study [46] in patients with megaesophagus, there is an adequate correlation between the **endoscopic** appearance of the **oesophageal mucosa** and the degree of esophagitis objectified by biopsy. They proposed the classification shown in Table 18.

Finally, although it is not a specific classification for Chagas disease, the latest update of the **Chicago classification** of oesophageal motility disorders [47] is included, because it is often referred to in Chagas disease studies, Table 19. It describes the diagnostic criteria for motility

**Table 19. Chicago v4.0 Classification and Definition of Manometric Disorders [47].**

| Classification | Disorder | Definition |
|---|---|---|
| Disorders of EGJ Outflow | Type I Achalasia | Abnormal median IRP & 100% failed peristalsis |
| | Type II Achalasia | Abnormal median IRP, 100% failed peristalsis, & 20% swallows with panesophageal pressurization |
| | Type III Achalasia | Abnormal median IRP & 20% swallows with premature/spastic contraction and no evidence of peristalsis |
| | EGJ Outflow Obstruction | Abnormal median IRP (supine **and** upright), 20% elevated intrabolus pressure (supine), and not meeting criteria for achalasia |
| Disorders of Peristalsis | Absent contractility | Normal median IRP (supine and upright) & 100% failed peristalsis |
| | Distal oesophageal spasm | Normal median IRP & 20% swallows with premature/spastic contraction |
| | Hypercontractile Oesophagus | Normal median IRP & 20% hypercontractile swallows |
| | Ineffective Oesophageal Motility | Normal median IRP, with >70% ineffective swallows or 50% failed peristalsis |

Integrated relaxation pressure (IRP); esophagogastric junction (EGJ)

**Table 20. Diagnostic techniques and variables utilized to assess and classify digestive involvement in CD.**

| Diagnostic technique/s | Variables |
| --- | --- |
| Esophagogram | Oesophageal calibre and morphology, contrast retention degree |
| Oesophageal manometry | Contractility and latency of the oesophageal body. Integrated Relaxation Pressure (IRP) |
| Barium enema | Colonic calibre and morphology. |
| Endoscopy | Macroscopic characteristics of the oesophageal mucosa |
| Biopsy | Microscopic characteristics of the oesophageal mucosa |

disorders present in CD as well as in other diseases, and is the reference classification for manometric studies in patients with CD.

A summary of diagnostic techniques utilized to assess and classify digestive involvement in CD can be found in Table 20.

## Uptake measurement

Tables 21 and 22 show the number of citations for each individual article in Google Scholar, PubMed and Dimensions, as well as the average number of citations for each of the available sources. If we exclude the citations of the Chicago generalist classification, the most cited CD digestive classification is that of Rezende, considering that both Brazilian consensuses incorporate it without modifications. Similarly, with the exception of the general classification of cardiomyopathies, the most commonly cited classification of CCD is the Rassi classification. The next most cited classification would be that of the second Brazilian consensus, which could be added to the citations of the first, since there are no significant changes. Thenceforth is the very similar classification found in the Latin American guidelines followed by Dr. Chagas' classic paper, and the Kuschnir classification. For the purpose of this paper, we exclude the

**Table 21. Uptake measurement of Digestive classifications by citations.**

| year | Title | #citations PubMed | #citations Dimensions | #citations Google Scholar | Mean of available sources |
| --- | --- | --- | --- | --- | --- |
| 1974 [28] | Clinical aspects of Chagas Disease | n/a | n/a | n/a | n/a |
| 2021 [48] | New Classification for Oesophageal Motility Disorders (Chicago Classification Version 4.0 | 130 | 377 | 434 | 313,7 |
| 2016 [12] | 2 nd Brazilian Consensus on Chagas Disease, 2015. | 117 | 211 | 287 | 205 |
| 2005 [39] | Brazilian Consensus on Chagas Disease | n/a | n/a | 59 | 59 |
| 1960 [15] | Clinical and radiological aspects of aperistalsis of the oesophagus | 30 | 50 | 47 | 42,3 |
| 2006 [41] | Clinical forms of *Trypanosoma cruzi* infected individuals in the chronic phase of Chagas disease in Puebla, Mexico | 14 | 26 | 58 | 32,7 |
| 2003 [45] | Proposed classification of chagasic megacolon by opaque enema | n/a | 7 | 10 | 8,5 |
| 1979 [29] | A review of the anatomo-clinical classification of Chagas' disease | 0 | 2 | 0 | 0,7 |
| 2012 [46] | Endoscopic classifications of oesophageal changes in chagasic megaesophagus | n/a | 0 | 0 | 0 |

**Table 22. Uptake measurement of Cardiologic classifications by citations.**

| year | Title | #citations PubMed | #citations Dimensions | #citations Google Scholar | Mean of available sources |
|---|---|---|---|---|---|
| 1967 [26] & 1968 [27] | The various types of chagasic myocarditis 1 & 2 | n/a | n/a | n/a | n/a |
| 1974 [28] | Clinical aspects of Chagas Disease | n/a | n/a | n/a | n/a |
| 1995 [35] | Clinical Classification of Chronic Chagasic Cardiomyopathy. XXII Congress of Cardiology International Symposium on Chagas Disease | n/a | n/a | n/a | n/a |
| 1996 [36] | Report of the 1995 World Health Organization/International Society and Federation of Cardiology Task Force on the Definition and Classification of cardiomyopathies | 716 | 2.783 | 4.433 | 2.644 |
| 2010 [14] | Chagas Disease | 881 | 1.806 | 2.944 | 1.877 |
| 2016 [12] | 2 nd Brazilian Consensus on Chagas Disease, 2015. | 117 | 211 | 287 | 205 |
| 2011 [13] | I Latin American guidelines for the diagnosis and treatment of Chagas' heart disease | 65 | 160 | 212 | 145,7 |
| 1916 [24] | Pathogenic processes of American trypanosomiasis | n/a | 78 | 164 | 121 |
| 1985 [33] | Life expectancy analysis in patients with Chagas' disease: prognosis after one decade (1973–1983) | 28 | 95 | 205 | 109,3 |
| 1985 [11] | Evaluation of Cardiac Function by Radioisotopic Angiography, in Patients with Chronic Chagas Cardiopathy | 48 | 95 | 111 | 84,7 |
| 1982 [10] | Left ventricular cineangiography in Chagas' disease: Detection of early myocardial damage | 13 | 76 | 14 | 76,3 |
| 1987 [34] | Clinical, histochemical, and ultrastructural correlation in septal endomyocardial biopsies from chronic chagasic patients: Detection of early myocardial damage | 17 | 73 | 120 | 70 |
| 2005 [39] | Brazilian Consensus on Chagas Disease | n/a | n/a | 59 | 59 |
| 1966 [25] | Clinical and epidemiological study of chronic heart involvement in Chagas | 9 | 38 | 102 | 49,7 |
| 1982 [30] | Electrocardiographic classification and abbreviated lead system for population-based studies of Chagas' disease | 8 | 38 | 65 | 37 |
| 2006 [41] | Clinical forms of *Trypanosoma cruzi* infected individuals in the chronic phase of Chagas disease in Puebla, Mexico | 14 | 26 | 58 | 32,7 |
| 2005 [38] | Application of the new classification of heart failure (ACC/AHA) in chronic chagasic heart disease: critical analysis of survival curves. | n/a | n/a | 32 | 32 |
| 2008 [43] | Prognostic value of natriuretic peptides in Chagas' disease: a 3-year follow-up investigation | 8 | 23 | 33 | 21,3 |

(*Continued*)

**Table 22.** (Continued)

| year | Title | #citations PubMed | #citations Dimensions | #citations Google Scholar | Mean of available sources |
|---|---|---|---|---|---|
| 2007 [42] | Assessment of myocardial damage in chronic chagasic patients using QRS Slopes | n/a | 5 | 7 | 6 |
| 1979 [29] | A review of the anatomo-clinical classification of Chagas' disease | 0 | 2 | 0 | 0,7 |

classification published in 2023 [44] due to insufficient elapsed time since its publication, rendering the measurement criteria inapplicable.

Among the 52 clinical trials for CD published on clinicaltrials.gov, 3 were excluded because they were conducted exclusively in the paediatric population. Of the remaining 49, 24 specified some type of classification to be considered in their inclusion criteria and/or in the corresponding publication as shown in Table 23. These classifications, with the exception of generic clinical involvement, which also includes digestive disease, largely refer to cardiac involvement. Most do so in relation to the clinical expression of heart failure (either through its presence or absence, NHYA classification or BNP levels) or through alterations in cardiological tests. Only six studies follow proper classifications of CD, Kuschnir being the most common.

The uptake pattern is similar in clinical regulations and guidelines. The PAHO guidelines, as well as most of the national guidelines of endemic countries, do not include a classification that supports the indication of treatment for patients with no or mild organ involvement. Among those that do, the most popular classification was that of Kuschnir: Argentina (2018), Mexico (2019) and Paraguay (2021). Other classifications are Los Andes in Colombia (2013) and Venezuela (2014), and NYHA in Mexico (2019). In the rest of endemic countries, there are either no clinical guidelines or those that exist do not use any specific cardiological or digestive classification.

**Table 23. Classifications used in Clinical Trials.**

| Criteria | Number of studies |
|---|---|
| None | 25 |
| NYHA | 5 |
| LVEF | 5 |
| **Kuschnir | 4 |
| Abnormal ECG | 4 |
| Clinical Heart Failure | 4 |
| *Clinical disease | 3 |
| BNP levels | 2 |
| Holter | 1 |
| **Brazilian consensus | 1 |
| **Latin-American Guidelines | 1 |

*Includes digestive and/or cardiologic.

**Actual CD classifications.

## Discussion

In the realm of Chagas disease (CD) research, a notable absence persists: the lack of a universally accepted clinical classification system for chronic cardiac involvement, both in the context of clinical trials and within current clinical guidelines. Commonly utilized parameters for this purpose encompass clinical assessments, chiefly reliant on the NYHA scale, electrocardiographic recordings, and evaluations of ventricular function. Notably, the presence of clinical heart failure, particularly manifesting as an NYHA score exceeding I, indicates moderate myocardial involvement, carrying adverse prognostic implications. Consequently, the most precise prognostic classifications judiciously exclude patients displaying heart failure symptoms from the initial categories of cardiac involvement or the indeterminate form. Notably, criteria for electrocardiogram (ECG) or Holter abnormalities exhibit substantial variability across different classification systems. In earlier iterations, invasive techniques were the norm for assessing ventricular function. However, a noteworthy shift transpired post-2005 [38], with the advent of trans-thoracic echocardiography (TTE) as the preferred modality for classifying cardiac involvement in Chagas disease. The subsequent revision in 2007 [49], while not entirely novel, marked a pivotal juncture in the field. This revision significantly substituted ventriculography with TTE and meticulously documented its equivalent prognostic value. This compelling evidence underscores the inclusion of TTE in the evaluation of individuals with *T.cruzi* infection, given that certain initial changes may elude detection via ECG alone [50]. It is crucial to acknowledge that the availability of TTE is not be uniform across endemic regions, and some efforts have been made to improve clinical management when TTE is not available [51]. Nevertheless, every effort should be made to ensure accessibility, given its profound implications for patient monitoring and management [52].

One of the primary functions of these classifications is to delineate patient groups with analogous clinical profiles, thereby facilitating prognostic assessments and informed clinical decision-making. Many variables featured in the clinical classifications outlined in this paper are well-established prognostic indicators. Nevertheless, it is essential to recognize the existence of numerous prognostic variables not encompassed by current classification systems [53]. This notably includes up to three TTE parameters, beyond the widely acknowledged left ventricular ejection fraction (LVEF), that possess prognostic significance: right ventricular function, left atrial volume, and the E/e' ratio. Additionally, other parameters assessable via ECG or Holter monitoring, such as the presence of non-sustained ventricular tachycardia (NSVT) or T-wave variability, contribute valuable prognostic insights. Several non-standard tests, including blood levels of B-type natriuretic peptide (BNP), signal-averaged QRS duration, exercise testing, and specific drug treatments, also exhibit prognostic utility.

Turning our attention to digestive classifications, it is noteworthy that these systems currently lack the inclusion of prognostic indicators during their development or monitoring phases. In cases where esophageal involvement presents as achalasia, manometry may complement the diagnostic approach. In the context of gastrointestinal involvement, Rezende's classifications for the esophagus and the presence or absence of megacolon enjoy widespread acceptance. These classifications primarily hinge on anatomical criteria, primarily employing barium contrast studies. While the role of endoscopy [46] in staging remains somewhat ambiguous, its utility may prove pivotal depending on the clinical presentation, as may ph-metry [48].

Regarding colonic involvement, the primary reference remains the presence or absence of megacolon as observed in barium studies. Attempts to establish a comprehensive staging system have yielded limited success [45], likely attributable to the absence of a direct correlation

between anatomical involvement extent and clinical impact. Consequently, its utility for decision-making purposes is notably limited.

Therefore, our study underscores the conspicuous absence of a robust, globally accepted classification for cardiac and digestive involvement in Chagas disease. This absence carries multifaceted implications, both at the individual and global levels. Individually, decisions regarding the administration of trypanocide treatment often hinge upon the presence or absence of organ involvement [54], yet the specific variables or classifications to consider remain elusive. This inconsistency poses substantial challenges at the clinical level, impacting decision-making and the universality of criteria for clinical trials and the licensing of new drugs. On a global scale, this void impedes meaningful comparisons among different Chagas disease studies and the establishment of a reliable registry of organ involvement.

In the realm of clinical trials, significant room for improvement exists, as only a few incorporate any organ involvement criteria into their protocols. Among those that do, the criteria often revolve around clinical alterations or complementary tests, rather than relying on an actual CD classification. Moreover, none of the international guidelines provide sufficient detail regarding the classification(s) to employ in clinical decision-making.

To our knowledge, this is the first comprehensive review of the diverse classifications pertaining to chronic organ involvement in Chagas disease. Employing a systematic search methodology to evaluate all historical CD classifications signifies a rigorous and exhaustive approach. Furthermore, the incorporation of languages beyond English, notably Spanish and Portuguese, enhances the breadth of our findings. Nevertheless, as with any systematic search, the possibility of overlooking pertinent studies cannot be entirely ruled out. To mitigate this potential, a meticulous examination of bibliographic citations from each publication was conducted, supplemented by additional references where applicable. Notably, despite our best efforts, certain publications remained inaccessible, even following attempts to contact authors and their affiliated institutions.

The absence of a standardized metric for assessing the acceptability of diverse classifications poses a methodological challenge in our study [55]. Our approach centres on evaluating their incorporation into clinical trial protocols and guidelines, supplemented by various citation metrics. These measures serve as pragmatic proxies for assessing the practical relevance of these classifications within clinical contexts.

Hence, both from clinical and epidemiological standpoints, the imperative arises for the development of a universally accepted CD classification system, one possessing robust prognostic value to effectively guide clinical strategies for patients with *T. cruzi* infection. Such a classification would also facilitate comparisons across clinical trials and the licensing of new drugs.

This work represents an initial step in this direction, encompassing a comprehensive review of all existing classifications in Chagas disease. The subsequent phase entails the creation of an exhaustive variable dictionary that will enable the clinical classification of CD patients, leveraging clinical manifestations and results from various complementary tests. This endeavour will facilitate cross-population comparisons if clinical manifestations and complementary test results are available for each patient, constituting a pivotal stride toward identifying precise prognostic parameters and establishing a minimum data set requisite for a fitting CD classification, tailored to the test availability in both endemic and non-endemic regions.

## Supporting information

**S1 Table. Search engine.**
(DOCX)

**S2 Table. Chagas Classifications.**
(DOCX)

## Author Contributions

**Conceptualization:** Irene Losada Galván, Joaquim Gascón, Maria-Jesus Pinazo.

**Data curation:** Irene Losada Galván.

**Formal analysis:** Irene Losada Galván.

**Funding acquisition:** Irene Losada Galván, Joaquim Gascón, Maria-Jesus Pinazo.

**Investigation:** Irene Losada Galván, Maria-Jesus Pinazo.

**Methodology:** Irene Losada Galván, Maria-Jesus Pinazo.

**Supervision:** Joaquim Gascón, Maria-Jesus Pinazo.

**Validation:** Maria-Jesus Pinazo.

**Visualization:** Irene Losada Galván, Sergi Sanz.

**Writing – original draft:** Irene Losada Galván.

**Writing – review & editing:** Irene Losada Galván, Magdalena García, Alejandro Marcel Hasslocher-Moreno, Ariadna Ortiga, Sergi Sanz, Israel Molina, Joaquim Gascón, Maria-Jesus Pinazo.

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
