## [Decision Letter · Decision Letter 0]

6 Mar 2024

Dear Ms Losada Galván,

Thank you very much for submitting your manuscript "How do we classify organ involvement in Chagas disease? A Systematic Review of Organ Involvement Since 1909, Highlighting the Urgent Need for a Universal Classification System in Chronic Chagas Disease" for consideration at PLOS Neglected Tropical Diseases. As with all papers reviewed by the journal, your manuscript was reviewed by members of the editorial board and by several independent reviewers. In light of the reviews (below this email), we would like to invite the resubmission of a significantly-revised version that takes into account the reviewers' comments. 

We cannot make any decision about publication until we have seen the revised manuscript and your response to the reviewers' comments. Your revised manuscript is also likely to be sent to reviewers for further evaluation.

Sincerely,

Walderez O. Dutra, PhD.

Section Editor

Walderez Dutra

Section Editor

Reviewer's Responses to Questions

**Key Review Criteria Required for Acceptance?**

**Methods**

-Are the objectives of the study clearly articulated with a clear testable hypothesis stated?

-Is the study design appropriate to address the stated objectives?

-Is the population clearly described and appropriate for the hypothesis being tested?

-Is the sample size sufficient to ensure adequate power to address the hypothesis being tested?

-Were correct statistical analysis used to support conclusions?

-Are there concerns about ethical or regulatory requirements being met?

Reviewer #1: Yes, goals clearly and objectively defined, despite not having a specific hypothesis being tested. 

Study design quite appropriate, by using systematic revision, without resource to statistical tools because not warranted.

Articles pertaining to the populations studies not entirely appropriate, as detailed below.

Neither sample size nor statistical analysis are applicable to the subject.

No concerns about regulatory or ethical issues.

Reviewer #2: see comments

**Results**

-Does the analysis presented match the analysis plan?

-Are the results clearly and completely presented?

-Are the figures (Tables, Images) of sufficient quality for clarity?

Reviewer #1: I have some remarks regarding these aspects. Please see below.

Reviewer #2: see comments

**Conclusions**

-Are the conclusions supported by the data presented?

-Are the limitations of analysis clearly described?

-Do the authors discuss how these data can be helpful to advance our understanding of the topic under study?

-Is public health relevance addressed?

Reviewer #1: For the conclusions topic I also have some points to be considered by the Authors. Please see belox.

Reviewer #2: see comments

**Editorial and Data Presentation Modifications?**

Reviewer #1: I would recommend minor revision by the authors, regarding a few points:

1 - In the Introduction section, it is stated that “Predicting the individual probability of disease progression remains a challenge, as there are currently no reliable tools available for this purpose”. It would be relevant to emphasize that although there is no clinical or laboratory predictor of how the disease will evolve in a specific individual, the mean probability of progression from the indeterminate form to the cardiomyopathy form has been recently defined by a systematic review and subsequent metanalysis. JAMA Netw Open. 2020 Aug 3;3(8):e2015072. Moreover, the mortality risk in patients already running through the cardiomyopathy form was also assessed using similar tools. Chadalawada et al. ESC Heart Fail. 2021 Dec;8(6):5466-5481.

2 - In the same section it is stated that “There are several clinical classifications for the visceral involvement of CD, but four of

them stand out, all staging Chagas disease with a focus on heart disease: Modified Los Andes (10), Kuschnir (11), 2nd Brazilian consensus (12) and I Latin American guidelines (13). All these classifications, attempt to stratify patient’s prognosis based on different information such as ECG, chest X-ray, echocardiogram, and clinical heart failure symptoms for cardiomyopathy, among others”. 

However, another classification, that of Rassi et al in The Lancet 2010, would merit to be quoted as such, according to the various aspects as analyzed later by the authors, in the text of their manuscript. I would suggest that the authors acknowledge that fact and possibly have a more critical take on the merits and limitations of each classification. 

3 - The authors also state that “Following the classification of chronic heart failure (CHF) published in 2001 by the

ACC/AHA(45), from which CD was excluded, a 2005 study(35) aimed to evaluate the prognostic performance of this classification applied to a cohort of patients with CD, Table 8. In a 5-year follow-up, no progression to Chronic Chagas Cardiomyopathy (CCC) was observed among patients who had a normal ECG at baseline. Therefore, only patients with an abnormal ECG at baseline were included. Patients were divided into 4 groups according to the results of ECG, echocardiogram, and clinical symptoms of heart failure. Survival curves showed a large difference between groups B and C, so group B was subdivided according to LVEF measured by echocardiography; this was the first classification to include a non-invasive measurement of ventricular capacity in Chagas.” This is not correct, Kuschnir used RNA to measure EF non-invasively, the method only requires an intravenous injection of the radionuclide agent, not intracardiac contrast ventriculography. In fact, the RNA should be considered the gold standard for measurement of both right and left ventricular ejection fraction, because it is quite more accurate than the echocardiographic technique. I also suggest the replacement of the word "capacity" by "systolic function" in the context.

3 - On what basis the review paper by Bern C et al published in 2007 by JAMA was excluded from the present review, by Losada et al? In that paper, following a systematic review of evidence, a detailed diagnostic basis for detecting the absence or presence of heart and digestive organ involvement is presented and also is used for prognostic and therapeutic measures in people with Chagas disease. 

4 - Another striking oversight by Losada et al efers to the AHA and InterAmerican Society of Cardiology statement published in 2018 in Circulation by Pereira Nunes et al, summarizing the most updated information on diagnosis and screening of patients after T, cruzi infection, focusing primarily on its cardiovascular aspects. Is the oversight justifiable?

5 - A recent e-book edited by two of the coauthors of the systematic review (MJPD and JG) - https://doi.org/10.1007/978-3-030-44054-1- included what is probably the most accurate and updated take on the esophageal and colon complications caused by Chagas disease. In the chapter entitled “Chagas Disease: An Unknown and Neglected Disease” results of studies for characterization of esophageal disorders in T. cruzi-infected individuals were reported in endemic and non-endemic countries, as well as the prevalence and severity of colonic involvement by de Oliveira RB et al. Could the authors of the manuscript reconsider and add that relevant information content?

Reviewer #2: see comments

**Summary and General Comments**

Reviewer #1: It is unfortunate that the recently published Guideline of the Brazilian Society of Cardiology on the Diagnosis and Treatment of Patients with Cardiomyopathy of Chagas Disease – Arq Bras Cardiol 2023 has not been included in the systematic methodology review carried out by the authors of the manuscript. Notwithstanding the circumstance that the whole guideline publication appeared only in June 2023, the Authors stated in their Abstract that “We searched relevant electronic medical databases from their inception dates to July 2023”. Also, an executive summary of the Guideline had already been published as a preprint in the Arquivos Brasileiros de Cardiologia, by December 2022. In addition, one of the coauthors of the manuscript was an active collaborator for the elaboration of the Guideline, rendering this oversight even more surprising and regretful.

Although the overall paper, for the purposes of the authors, stands well without that last guideline, I would suggest that they could reconsider, and give the manuscript the chance of being enriched by the addition of the data conveyed by this last guideline. 

In addition to that general suggestion, I have some few relevant remarks to be considered by the authors, should they agree to perform a major revision: 

In essence, the authors documented the lack of generally accepted criteria for the diagnosis and prognosis of organ involvement in Chagas disease, especially in regard to the cardiomyopathy form. They call for a standardization of criteria that does not exist in the realm of the various classification and other related papers they systematically reviewed. I am afraid that when published, as it deserves to be, this paper will sound rather anodyne and missing the opportunity to perform a more critical review of the criteria they identified in the classifications analyzed. I know that the review aimed solely at reporting on the diverse criteria used in various classifications of organic involvement due to Chagas disease. That goal was indeed accomplished. However, without a more critical appraisal of the data, the highly categorized authors of the investigation may have missed a good opportunity of a real advance in the field. Ideally, their team could have come up with some proposition that would reflect a synthesis of the heterogenous data assembled and perhaps become more widely accepted. 

Nevertheless, provided that Losada and collaborators include in their revision the recent guideline of the Brazilian Society of Cardiology just published, it is quite possible that it could constitute a starting point for the proposal here suggested. In fact the new Brazilian guideline compiled data from a more extensive array of documents than that surveyed by Losada et al. A thorough review of diagnostic methods for detection of structural and functional cardiac abnormalities was presented, with a heart failure classification dedicated to CD peculiarities added. Detection of rhythm derangements and thromboembolic complications was also fully addressed. Also, the guideline extensively dwelled on a critical appraisal of risk stratification methods for prognosis of patients with Chagas disease.

Reviewer #2: The MS intitled “How do we classify organ involvement in Chagas disease? A Systematic Review of Organ Involvement Since 1909, Highlighting the Urgent Need for a Universal Classification System in Chronic Chagas Disease” by Losada I et al., submitted to PLoSNTD as a Research Article, cannot be accepted in the present format and content, based on the following considerations:

1- The title confuses and misleads readers. As Chagas disease is a complex, systemic disease, that may have outcome influenced by non-genetic factors of host (as nutritional, pre-immunological status) and genetic factors of host and parasite, is there a need for a universal classification of Chagas disease? If so, it is not this “Systematic Review” that highlights it, but rather the current needs of patients, the barriers faced by professionals in the healthcare system and the people who spend effort in clinical trials. 

2- The classification of the MS as a “Research Article” is not correct, as it is a review on a subject or could be a “Political Platform”. In fact, this last format may call attention to the need for an international effort focused on Chagas disease as a systemic disorder with different clinical forms. Also, considering the contribution of aspects of the host (genetic, nutritional condition) and the infecting parasite population to clinical outcomes, there are aspects to be completely identified before classification. I agree that Chagas disease deserves greater attention to clinical outcomes and clinical classification (cardiac, digestive, cardi-digestive, neurological and even behavioral, as indicated more recently; for example, see Silva et al., 2020, focused on depression) to allow proper diagnosis and care. Thus, a more political perspective may add call more attention to this point.

3- Carlos Chagas paid attention to several clinical aspects of the acute phase of the disease he described: the common symptoms of acute infection; a systemic disease (the mistake with goiter, has been already overcome, as a consensus); the cardiac acute form; and the acute nervous form (particularly, in children). All these are crucial contributions of Chagas, 1916. But the authors, more fixed to timeline, jumped to an article describing the cardiac form in the chronic phase (not even the first one to describe the heart disease). The clinical points of the acute phase, crucial for diagnosis and care, were not dissected. It is a point to be revised.

4- Considering the “proposed aim” “organ involvement”, the neurological form (neurobehavioral form), mostly as a sequelae of stroke, is often the first clinical sign of the chronic Chagas disease, crucial for diagnosis in people under 40’, as it is a crucial point to be considered in Latin America (and, particularly Latin American individuals residing in North America, Europe and Oceania). This point has not been properly discussed. 

5- The “timeline format” should be summarized. Although interesting, it is not the crucial point of this MS (for example, line 149 to 270). The authors should focus on the consensus (or lack of consensus) of the last two decades. 

6- The format of this revision, the selected words, the presentation of the MS is a pitfall, considering the so good group of researchers involved in this task and the rele

---

## [Decision Letter · Decision Letter 1]

12 Jul 2024

Dear Ms Losada Galván,

We are pleased to inform you that your manuscript 'How do we classify organ involvement in Chagas disease? A Systematic Review of Organ Involvement Since 1909, Highlighting the Urgent Need for a Universal Classification System in Chronic Chagas Disease' has been provisionally accepted for publication in PLOS Neglected Tropical Diseases.

Best regards,

Abhay R Satoskar

Section Editor

Walderez Dutra

Section Editor

Reviewer's Responses to Questions

**Key Review Criteria Required for Acceptance?**

**Methods**

-Are the objectives of the study clearly articulated with a clear testable hypothesis stated?

-Is the study design appropriate to address the stated objectives?

-Is the population clearly described and appropriate for the hypothesis being tested?

-Is the sample size sufficient to ensure adequate power to address the hypothesis being tested?

-Were correct statistical analysis used to support conclusions?

-Are there concerns about ethical or regulatory requirements being met?

Reviewer #2: No alteration was detected.

**Results**

-Does the analysis presented match the analysis plan?

-Are the results clearly and completely presented?

-Are the figures (Tables, Images) of sufficient quality for clarity?

Reviewer #2: Results were partially improved.

**Conclusions**

-Are the conclusions supported by the data presented?

-Are the limitations of analysis clearly described?

-Do the authors discuss how these data can be helpful to advance our understanding of the topic under study?

-Is public health relevance addressed?

Reviewer #2: Yes

**Editorial and Data Presentation Modifications?**

Reviewer #2: Although the authors respectfully disagree with most of our comments, the text would benefit from accepting a few more suggestions (as already noted in the current version).

I believe that reading this article can help young doctors and/or researchers to look at the challenge of clinical classification of Chagas disease and its importance for the clinical management of people affected by this disease. Therefore, I consider that the publication of this article can be useful, particularly as a stimulus to the development of critical thinking, which I hope will be expressed in other articles that will follow. Furthermore, it may bring engagement in the search for a solution to this problem. Thus, the article may be accepted after grammatical and spelling review.

**Summary and General Comments**

Reviewer #2: Text would benefit from accepting a few more suggestions (as already noted in the current version).

The article may be accepted after grammatical and spelling review.

PLOS authors have the option to publish the peer review history of their article (what does this mean?). If published, this will include your full peer review and any attached files.

Reviewer #2: No

---

## [Editor Report · Acceptance letter]

31 Jul 2024

Dear Ms Losada Galván,

We are delighted to inform you that your manuscript, "How do we classify organ involvement in Chagas disease? A Systematic Review of Organ Involvement Since 1909, Highlighting the Urgent Need for a Universal Classification System in Chronic Chagas Disease," has been formally accepted for publication in PLOS Neglected Tropical Diseases.

Best regards,

Shaden Kamhawi

co-Editor-in-Chief

Paul Brindley

co-Editor-in-Chief
